# Dengue virus IgG and neutralizing antibody titers measured with standard and mature viruses are protective

Camila D. Odio[1], Jedas Veronica Daag[2], Maria Vinna Crisostomo[2], Charlie J. Voirin [1], Ana Coello Escoto[1], Cameron Adams[3], Lindsay Dahora Hein[3], Rosemary A. Aogo [1], Patrick I. Mpingabo[1], Guillermo Raimundi Rodriguez[1], Saba Firdous[1], Maria Abad Fernandez[3], Laura J. White[3], Kristal An Agrupis[2], Jacqueline Deen [2], Aravinda M. de Silva [3], Michelle Ylade [2] ✉ & Leah C. Katzelnick [1] ✉

The standard dengue virus (DENV) neutralization assay inconsistently predicts dengue protection. We compare how IgG ELISA, envelope domain III (EDIII), or non-structural protein 1 (NS1) binding antibodies, and titers from plaque reduction neutralization tests (PRNTs) using standard and mature viruses are associated with dengue. The ELISA measures IgG antibodies that bind to inactivated DENV1-4. The EDIII and NS1 assays measure binding antibodies, and the PRNTs measure neutralizing antibodies to each specific DENV serotype. Healthy children ($n = 1206$) in Cebu, Philippines were followed for 5 years. ELISA IgG≥3 was associated with reduced dengue probability relative to naïve children (3% vs. 10%, $p = 0.007$). Serotype-specific antibodies binding EDIII or NS1 had no association with dengue risk. Standard virus PRNT geometric mean titers (GMT) > 200 and mature GMT > 100 were associated with reduced dengue disease overall ($p < 0.01$). High DENV2 and DENV3 titers against either standard or mature viruses protected against the matched serotype ($p < 0.01$). While 43% of dengue cases had standard virus PRNT titers>100, only 2% of cases had mature virus PRNT titers>100 ($p < 0.001$), indicating a lower, more consistent threshold for protection. These assays may serve as correlates of protection.

Dengue is a febrile illness caused by any one of four co-circulating viruses, dengue virus serotypes 1-4 (DENV1-4), with disease ranging from undifferentiated fever to severe manifestations including vascular leakage, shock, bleeding, and organ impairment[1]. An estimated 75% of infections are asymptomatic[2], and these inapparent infections likely contribute to viral spread. DENV1-4 is transmitted among humans by mosquitos in >100 tropical and subtropical countries. Globally, at least 2.5 billion people are at risk for dengue, and dengue is one of the few infectious diseases with a rising incidence in this century[3,4]. Between 1990 and 2019, the global number of dengue cases increased by 85% to 56.88 million in 2019[5]. The heavy healthcare burden and rising incidence have motivated dengue vaccine development

[1]Viral Epidemiology and Immunity Unit, Laboratory of Infectious Diseases, National Institute of Allergy and Infectious Diseases, National Institutes of Health, Bethesda, Maryland, USA. [2]Institute of Child Health and Human Development, National Institutes of Health, University of the Philippines Manila, Manila, Philippines. [3]Department of Microbiology and Immunology, University of North Carolina School of Medicine, Chapel Hill, North Carolina, USA. ✉e-mail: mcylade@up.edu.ph; leah.katzelnick@nih.gov

for decades, but these have provided incomplete protection, especially among individuals with no DENV antibodies at the time of vaccination[6,7].

An important limiting factor in the development of dengue vaccines is the lack of widely accepted correlates of protection. In the context of vaccines, correlates of protection are biomarkers that can be induced by immunization and reliably predict vaccine efficacy in preventing a clinical outcome[8,9]. Because correlates of protection are often easier and faster to measure than the desired clinical outcome, they can expedite trial results and vaccine approvals[10]. Once identified, regulators prefer that a correlate of protection have a single threshold that distinguishes those who get disease from those who do not get disease or even infection, but this determination can be complicated by heterogeneity in exposures and responses[11]. In addition, although an intervention meets the correlate of protection threshold at one-time point, threshold antibody levels may not persist due to waning immune responses. Thus, although a goal biomarker level is intuitive, a relative correlate of protection, where the probability of disease decreases gradually with increasing levels of the immune marker, maybe more realistic[11].

To establish a correlate of protection, a biological assay is often first identified as a correlate of risk in longitudinal cohort studies. Correlates of risk are biomarkers associated with both natural infection and disease outcomes, and it is important to distinguish between markers of immune protection versus those that simply predict the probability of exposure to the pathogen. Controlling for covariates that independently explain disease risk (e.g., age, sex, measures of local transmission intensity, etc.) helps build evidence that a correlate of risk could be a plausible correlate of protection[9].

For dengue vaccines, correlates of protection have not been widely accepted due to the inability to identify a single threshold, the complexities of the four interactive and immunomodulatory serotypes, and fundamental differences in vaccine responses between dengue seronegative and seropositive individuals. Numerous studies by Gilbert and colleagues have established that neutralizing antibody (nAb) titers measured using standard plaque reduction neutralization tests (PRNT) are correlates of protection in phase 3 dengue vaccine efficacy trials[12–14]. However, a single nAb threshold cannot be identified because even some individuals with exceptionally high titers (e.g., > 500) remain at risk of disease[13]. This contrasts with other vaccines against flaviviruses. For instance, a PRNT titer of 1:5 is protective against yellow fever virus infection and 1:10 against Japanese encephalitis virus infection, while 125 ELISA units are protective against tick-borne encephalitis virus infection[11]. Moreover, since DENV nAbs must be measured against all four serotypes, it is unclear whether correlates are different for each serotype or if a common correlate is feasible. Finally, correlates of protection against dengue may not be binary due to antibody-dependent enhancement (ADE), where low to intermediate levels of antibodies can increase the risk of severe disease compared to those with no antibodies or high antibodies.

Recent work indicates that antibody quality may be playing a critical yet underexplored role in dengue correlates. The validated (standard) PRNTs for measuring correlates of protection in dengue vaccine trials cannot distinguish between high-quality antibodies that are mechanistically protective and lower-quality antibodies that may increase the risk of severe disease. There is an emerging consensus that type-specific, or homotypic, nAbs induced by vaccines provide protective efficacy against dengue, and this population can be profiled using antibody depletion assays[15,16]. Cross-reactive antibodies target multiple serotypes and can either be low quality and associated with ADE or high quality and broadly neutralizing[17,18]. Assays that use mature rather than standard viruses to measure nAbs may better distinguish antibody quality and serve as more accurate correlates of protection, measuring both protective type-specific and cross-reactive antibodies[19,20].

Specifically, dengue virions are assembled as immature virions containing 60 copies of envelope (E) and pre-membrane (prM) heterotrimeric spikes on the surface. During virus egress from cells, prM is cleaved to generate an infectious mature particle containing 90 E homodimers that lie flat on the viral surface. The cleavage of prM is inefficient in laboratory cell lines, and thus, the standard virions used in the PRNT assay are heterogenous and display prM and fusion loop epitopes. Standard virions are sensitive to neutralization and/or enhancement by antibodies to fusion loop or prM epitopes, which are hidden or absent in fully mature virions[17,21]. While the processing and maturation state of DENV in humans remains to be well defined, a study has demonstrated that DENV1 in humans is fully mature and insensitive to fusion loop and prM antibodies[22]. Mature virions lack prM, resulting in only intermittent exposure of the fusion loop and a dimer configuration of the E protein that binds potent, type-specific antibodies. Compared to standard virions, an assay using mature virions is likely to be more selective, measure potent nAbs, and be less sensitive to neutralization or enhancement by fusion loop and prM antibodies[18].

Here, we evaluate multiple antibody measures as correlates of risk in the context of a natural infection cohort study in the Philippines. We examine a commercial IgG ELISA (PanBio; Brisbane, QLD), binding antibodies to E domain III (EDIII) or non-structural protein 1 (NS1) from each serotype, a PRNT assay using the reference strains recommended by the World Health Organization (WHO) and grown in standard laboratory conditions (standard), and a PRNT assay using circulating Southeast Asian strains (clinical) grown in conditions that induce mature virions as a more plausible 'mechanistic' correlates of risk. Our work compares multiple antibody markers as potential correlates of risk for use in future vaccine trials and for measuring protection and risk in natural dengue settings.

## Results
### Baseline characteristics
Children aged 9–14 years and residing in Bogo or Balamban, Cebu, Philippines, were recruited for a longitudinal observational study examining dengue risk among those who were eligible to receive CYD-TDV (Dengvaxia, Sanofi Pasteur)[23,24]. Participants provided verbal assent to the study, while written informed consent was provided by a parent or legal guardian. In total, the study followed $n = 1790$ children who received a single dose of the vaccine and $n = 1206$ children who were not vaccinated. The unvaccinated and vaccinated children had similar age, sex, and baseline serostatus, defined by testing baseline samples with the PanBio IgG ELISA and PRNT, as described previously[23,24] (Supplementary Table 1). The unvaccinated group was more likely to reside in Balamban (51% vs. 46%, $p = 0.004$) and had lower baseline dengue IgG ELISA values ($p = 0.02$). To assess the impact of natural immunity on the probability of dengue, only unvaccinated children were included in our analyses.

We evaluated whether baseline demographic characteristics and immune status were associated with the risk of symptomatic dengue in the unvaccinated children in the cohort. There were no differences in age at enrollment, sex, or residential site between those that developed symptomatic dengue (defined using the 2009 WHO dengue criteria as dengue without warning signs, DwoWS, dengue with warning signs, DWWS, and severe dengue) and those that did not (Table 1). Although most of the participants had multitypic immunity by standard PRNT, defined as neutralization of more than one serotype, the dengue attack rate was highest in the monotypic group who neutralized one serotype (9.6%), followed by the naïve (7.2%) and multitypic groups (3.7%) ($p = 0.005$). Those who developed symptomatic dengue had lower IgG ELISA values ($p = 0.036$). Of the 57 cases, there was no severe dengue, 32 had DWWS (DENV1, $n = 8/12$ cases; DENV2, $n = 9/17$; DENV3, $n = 12/22$; DENV4, $n = 3/5$). d One participant had DwoWS with both DENV1 and DENV3 detected by RT-PCR. There were

**Table 1 | Demographic characteristics and immune status of unvaccinated cohort participants**

| Characteristic | No Dengue (n = 1149)[a] | Symptomatic Dengue (n = 57)[a] | p-value[b] |
|---|---|---|---|
| Age | 10.92 (1.40) | 10.91 (1.15) | > 0.9 |
| Sex | – | – | > 0.9 |
| Female | 600 (52%) | 29 (51%) | – |
| Male | 549 (48%) | 28 (49%) | – |
| Site | – | – | 0.3 |
| Balamban | 593 (52%) | 25 (44%) | – |
| Bogo | 556 (48%) | 32 (56%) | – |
| Baseline Standard PRNT Serostatus | – | – | 0.005 |
| Naive | 128 (11%) | 10 (18%) | – |
| Monotypic | 113 (9.8%) | 12 (21%) | – |
| Multitypic | 908 (79%) | 35 (61%) | – |
| ELISA | 2.53 (1.07) | 2.20 (1.12) | 0.036 |

[a]Mean (SD); n (%), [b]Welch Two Sample t-test; Pearson's Chi-squared test, all tests are two-sided.

no differences in the demographics or baseline immune status between participants who developed DwoWS versus DWWS (Supplementary Table 2).

## Cross-reactive antibodies and the probability of symptomatic dengue

We next evaluated whether the probability of experiencing dengue (DwoWS and DWWS) could be predicted by total binding antibodies measured at baseline using an IgG ELISA and nAbs measured as the geometric mean of PRNT titers (GMT) against standard WHO DENV1-4 reference strains (standard) or mature clinical DENV1-4 isolates (mature). After adjusting for age, sex, and recruitment site, high total binding antibodies or nAb against standard or mature strains were associated with lower probabilities of dengue caused by any serotype as compared to the naïve group (Fig. 1). Specifically, the probability of dengue in the naïve group was 10% (95% CI: 5–20) versus 3% (2–6) among those with ELISA values ≥ 3 (p = 0.007). In the models examining standard GMT, the probability of dengue was 9% (4–17) in the naïve group versus 2% (1–4) among those with a standard GMT > 200 (p = 0.0009). With the mature PRNT, the probability of dengue was 12% (6–23) in the naïve group versus 1% (0–4) among those with mature

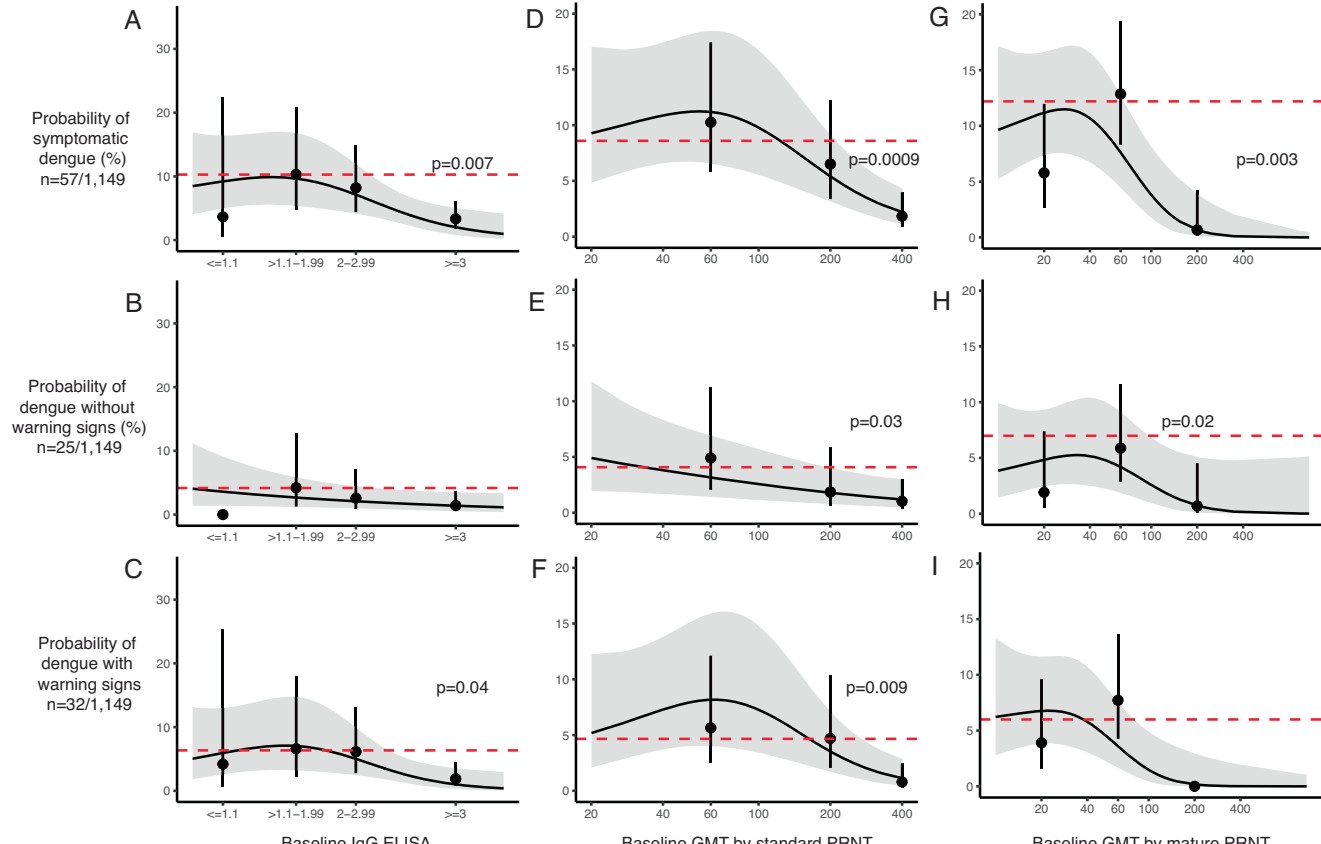

**Fig. 1 | Probability of dengue by baseline antibody measure.** The probability of disease is shown separately for baseline IgG ELISA (**A**–**C**) and baseline GMT measured by PRNT with standard reference (**D**–**F**) or mature clinical (**G**–**I**) strains. The probability of each disease outcome was modeled as a function of baseline antibody titer on both discrete and continuous scales. All continuous relationships were modeled using Poisson generalized additive models (continuous black line) with 95% confidence intervals (gray shading). Point estimates and confidence intervals correspond to predicted probabilities from logistic regression models. P-values were generated from the logistic regression models using two-sided tests comparing the dengue risk for each antibody bin with that in the naïve group. Only statistically significant p-values are shown. The dashed line indicates the disease probability in naïve individuals. DENV IgG ELISA was measured on all participants (n = 1,206), and GMTs were measured on random subsets using standard reference (n = 823) and mature clinical strains (n = 293). Inverse probability weighting was used to adjust for GMT subset size. All models were adjusted for age, sex, and enrollment site, and model estimates are shown for the average study participant (female, age 10, from Bogo). N for each row indicates the number of dengue cases and non-cases included in each analysis. Source data are provided as a Source Data file.

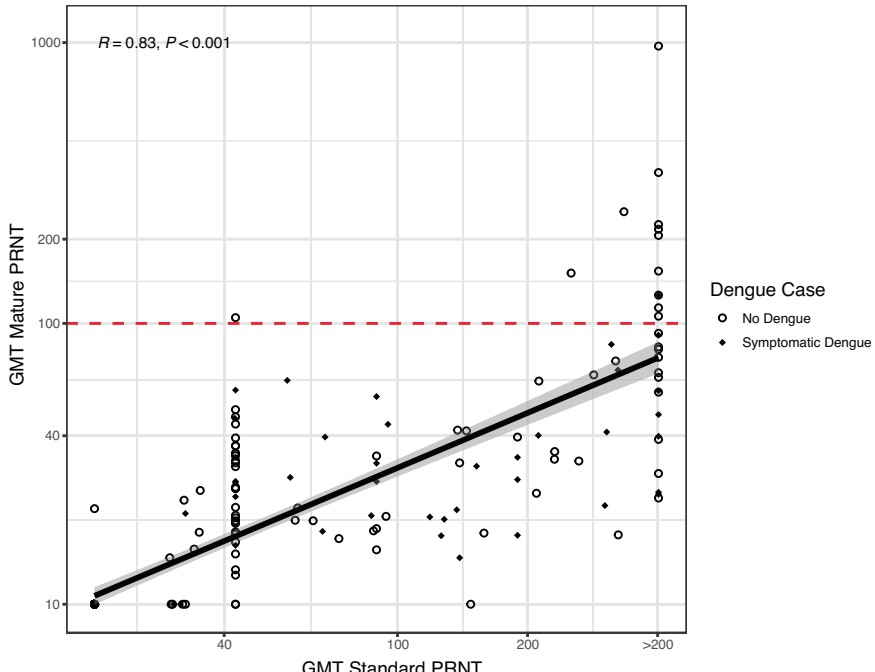

**Fig. 2 | Fitted linear regression line with 95% confidence intervals between GMTs as measured using PRNTs to standard reference and mature clinical strains.** GMTs are plotted on a log scale, and the correlation coefficient (R) was calculated using Pearson's test with 95% confidence intervals. Only those with no dengue (open circles) or dengue (black diamond) and GMTs measured by both the standard and mature PRNTs are included ($n = 256$). The dashed red line indicates GMT = 100 by mature assay. Source data are provided as a Source Data file.

GMT > 100 ($p = 0.003$). At lower antibody levels, none of the assays showed either enhancing or protective effects.

We also evaluated whether these same assays were associated with DwoWS or DWWS alone. Standard PRNT was associated with protection against both DWWS and DwoWS. IgG ELISA was associated with protection against DWWS, and the mature PRNT was associated with protection against DwoWS. Both assays also showed trends toward protection against DwoWS and DWWS, respectively. The association between dengue and high total binding antibodies and nAbs suggests that these antibody measures are correlates of risk.

To compare the antibodies measured by standard versus mature PRNT, we evaluated the association between the GMTs from these assays and found that they were highly correlated ($R = 0.83$). However, only 2% of cases occurred among those with mature GMT > 100, while 43% of cases occurred among those with standard GMT > 100 ($p < 0.001$) (Fig. 2). Notably, 29% of those with standard GMT > 200 had mature GMT > 100, and only 1% of those with standard GMT < 200 had mature GMT > 100. Thus, although correlated with the standard GMT, the mature GMT had a more consistent association with protection at the titers measured.

### Serotype-specific antibodies and risk of dengue

We next assessed both the GMT and titers to each serotype measured by standard and mature PRNT as predictors of symptomatic (over the 5-year follow-up period) and inapparent (over the first 1-2 years of follow-up) DENV infections caused by any serotype (Fig. 3). Notably, among individuals who did not have symptomatic dengue, inapparent infections occurred in 54% of naïve, 49% of monotypic, and 35% of multitypic individuals. Both standard and mature GMTs were strongly protective against symptomatic and inapparent dengue with similar reductions in the odds of dengue per $\log_{10}$ increase in GMT (odds ratios ranged from 0.29–0.55 [CIs: 0.14–0.76]). Interestingly, titers to each of the four serotypes were also associated with decreased odds of inapparent and symptomatic infection caused by any serotype, except for mature DENV2 titers, which did not protect against either infection.

The serotype-specific standard PRNT odds ratios ranged from 0.31–0.75 (0.20–0.96), and the protective mature PRNT odds ratios ranged from 0.29–0.64 (0.16–0.97) per $\log_{10}$ increase in titer. Thus, both the GMT and titers to each serotype show protective effects when measured against dengue caused by any serotype, especially when measured by the standard PRNT.

To evaluate if the standard PRNT may identify more cross-reactive antibodies as compared to the mature PRNT, we next analyzed disease risk individually by serotype for cases caused by DENV2 and DENV3, the most common serotypes to cause disease in the cohort. Titers measured by the mature PRNT had strong protective effects against the matched serotype (Fig. 4A, B). The odds of experiencing a DENV2 case were 0.13 (0.03–0.56) per $\log_{10}$ increase in mature DENV2 PRNT titer. Similarly, the odds of a DENV3 case were 0.12 (0.04–0.39) per $\log_{10}$ increase in mature DENV3 PRNT titer. The mature GMT did protect against DENV3 but only trended toward protection against DENV2, perhaps due to a smaller number of DENV2 cases. For the standard PRNT, a weaker effect size was observed for DENV2 titers and reduced odds of DENV2 case (odds ratio of 0.32 [0.15–0.72]), and the DENV4 titer and the GMT had some association with protection (odds ratio 0.38–0.4 [0.15–0.99]). Similarly, the standard titers to DENV3 protected against DENV3 case (odds ratio of 0.27 [0.13–0.58]), and the standard DENV1 titer and GMT were also associated with decreased odds of experiencing a DENV3 case (odds ratio 0.22–0.29 [0.10–0.69]). Thus, both standard and mature titers were highly protective against the matched serotype, but nAbs measured by mature PRNT appeared to have more serotype-specific protective effects. When visualizing the titer data directly, there was a difference in titer magnitude between cases and non-cases by both assays (Fig. 4C, D), but most of those with breakthrough cases had mature serotype matched titers below the assay limit of detection of 1:20, whereas a larger fraction of breakthrough cases had standard serotype matched titers above the assay limit of detection of 1:40 (mature: 7/38, 18%, standard: 27/45, 60%, two-sample test for equality of proportions, $p = 0.0003$).

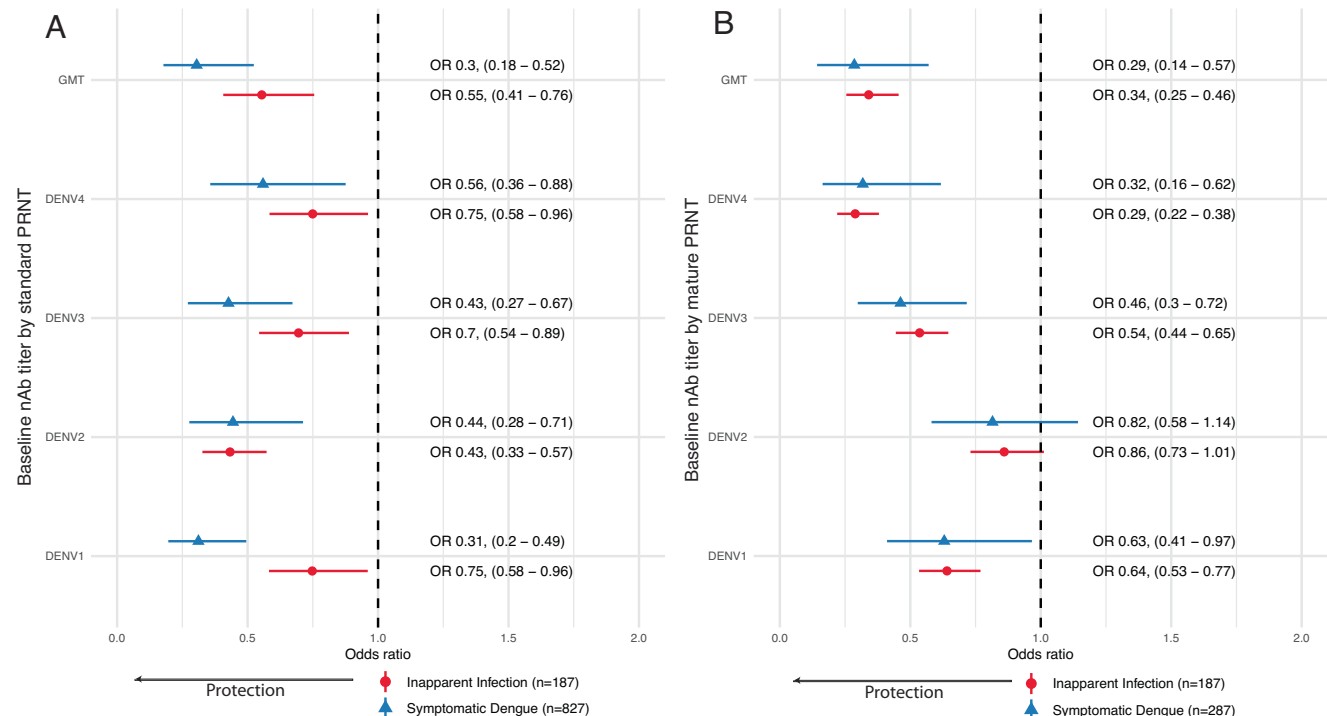

**Fig. 3 | Comparison of inapparent infection and symptomatic dengue to non-cases by baseline GMT and serotype-specific nAbs.** The odd ratios and 95% confidence intervals of inapparent infection (circles) and symptomatic dengue (triangle) by baseline GMT and serotype-specific nAbs measured with PRNT to standard reference (**A**) or mature clinical (**B**) strains. *N* = number of individuals evaluated with and without symptomatic or inapparent infection in each model. All analyses were logistic regression models adjusted for age, sex, and enrollment site, and inverse probability weighting was used to adjust for GMT and serotype-specific nAb subset sizes. Source data are provided as a Source Data file.

We then measured the correlations among titers with the expectation that related (cross-reactive) antibody populations would have stronger associations than distinct (more type-specific) populations. The strongest correlation was observed between the standard DENV1 and DENV3 assay titers ($r = 0.79$, Supplementary Fig. 1A). In contrast, the other standard DENV titers had weaker correlations ($r = 0.53–0.65$), and the mature DENV titers had the weakest correlations among serotypes ($r = 0.17–0.42$, Supplementary Fig. 1B). Standard serotype-specific titers had stronger correlations with standard GMT ($0.79$-$0.9$) than mature serotype-specific titers had with mature GMT ($0.58–0.77$).

Previous studies have shown that DIII of the envelope protein and NS1 protein have serotype-specific and cross-reactive epitopes. Particularly, EDIII has major type-specific epitopes on the lateral ridge and subdominant cross-reactive epitopes. We thus evaluated whether the magnitude of antibody binding to EDIII and NS1 from DENV2 and DENV3 predicted the risk of disease by the matched serotype. Neither EDIII nor NS1 binding antibodies was associated with risk (Supplementary Fig. 2).

**Antibody thresholds associated with protection against dengue**

To compare the quality of the correlates, we assessed the antibody thresholds associated with specific percent reductions in dengue disease compared to being naive (Table 2), similar to reporting vaccine efficacy associated with specific antibody titer quantities[13]. While only the highest ELISA values (4.5) and standard GMT (>200) were associated with a 70% disease reduction compared to the naïve group, a mature GMT of 114 was associated with 70% disease reduction. When serotype-specific protection was evaluated, titers measured by mature PRNT provided 70% protection at lower levels than the standard PRNT against DENV2 (40 vs. >200) or DENV3 (40 vs. 181). However, a 90% reduction in disease was only observed at a mature GMT of 344, meaning some individuals still experienced disease with highly mature

PRNT titers. Thus, serotype-matched mature titers had the strongest associations with protection at the lowest levels.

**Impact of strain and maturation state on titers**

To assess the impact of viral strain as compared to maturation state on nAb titers, a direct comparison of titers measured against standard and mature reference and clinical strains was performed (Fig. 5). Across the four strain and maturation state combinations, the mean GMT was highest against the reference standard virus (195), followed by the clinical standard virus (67). The lowest GMTs were observed against the mature clinical (53) and mature reference strains (43), and these were similar to each other, suggesting that the maturation state might be more important than strain in determining viral titers (Fig. 5A).

When stratified by serotype, the trends among strain and maturation states were more variable (Fig. 5B–E). Specifically, the reference standard viruses had the highest titers across serotypes, except against DENV3, where the titers measured by reference standard and reference mature viruses were similar (231 vs. 296, $p = $ NS). In contrast, the reference mature strain induced the lowest titers for DENV2. The clinical standard and mature viruses induced similar titers for all serotypes, except DENV2, where the clinical mature virus induced the lower titers (DENV2 clinical standard: 141 vs. clinical mature: 60, $p < 0.0001$). Thus, the maturation state may play a bigger role in determining titers for reference rather than clinical strains. In addition, the clinical mature strains consistently yielded lower titers across serotypes and may be the most selective assay for measuring antibody neutralization.

**Discussion**

We leveraged a large prospective, observational cohort to assess various antibody measures as potential correlates of risk for dengue. Total binding antibodies and both GMT and serotype-specific nAbs

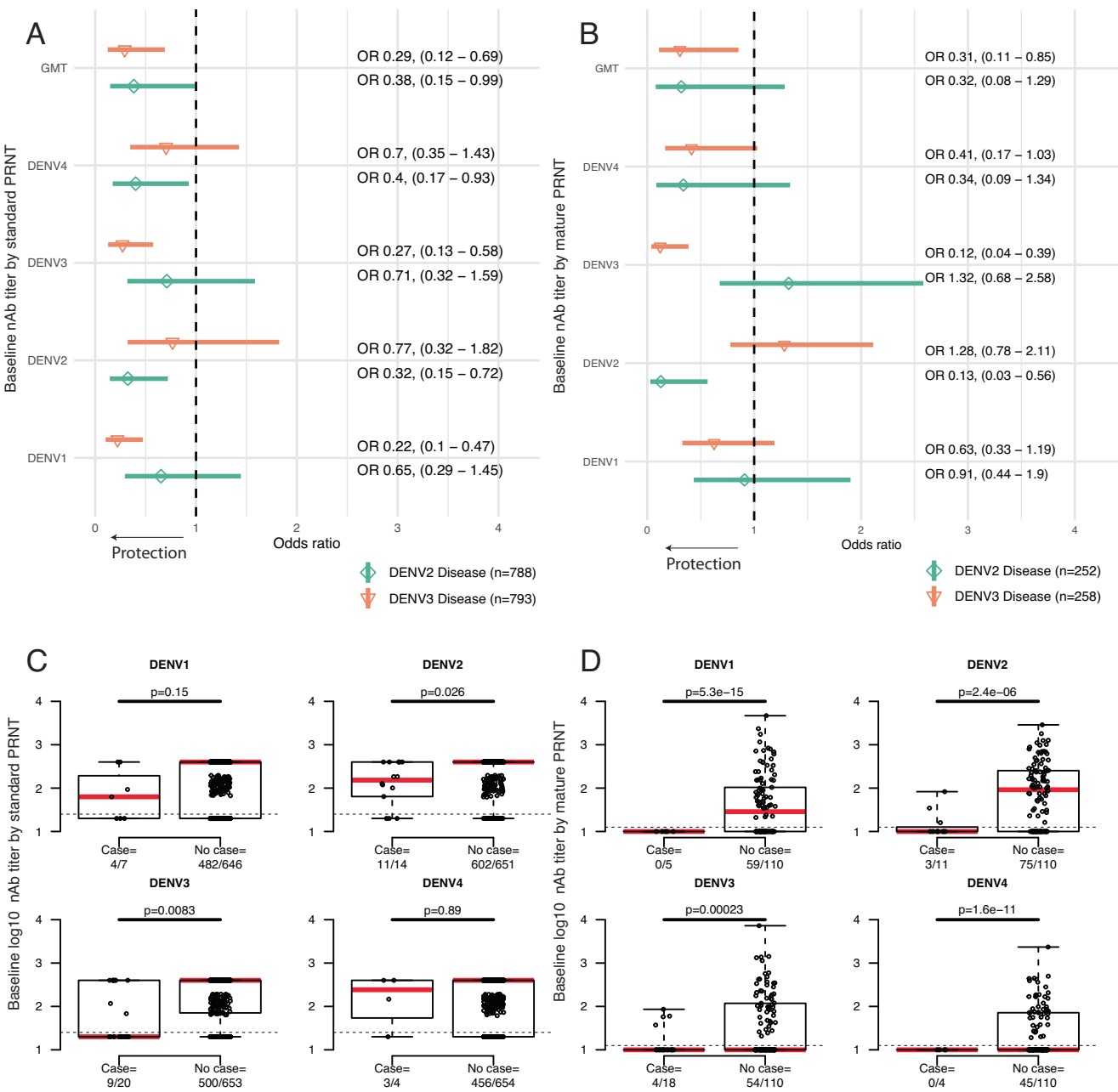

**Fig. 4 | The impact of standard and mature nAbs on cases caused by DENV2 and DENV3.** The odd ratios and 95% confidence intervals of experiencing a case of DENV2 (diamonds) or DENV3 (inverted triangles) by baseline titer were measured with standard reference (**A**) or mature clinical (**B**) strains. *N* = number of individuals with and without DENV2 or DENV3 disease evaluated in each model. All analyses were logistic regression models adjusted for age, sex, and enrollment site, and inverse probability weighting was used to adjust for GMT and nAb subset sizes. Individual-level data and distributions (medians in red, boxes show interquartile ranges, whiskers extend to show the full range of the data) of baseline titers for monotypic and multitypic immune participants, stratified by those who experienced a case caused by each serotype or did not experience a case, shown separately for standard reference (**C**) or mature clinical (**D**) neutralization assay results. A Welch two-sample, two-sided *t* test was used to compare differences in titer means between cases and non-cases for each analysis, with *p*-values reported. The number of individuals who experienced a case with that serotype or did not have a case are shown as the denominator under each boxplot, while those in each category with titers above the assay threshold (shown as a dotted line, 1:20 for the mature and 1:40 for the standard assay) are indicated in the numerator. Source data are provided as a Source Data file.

measured with standard and mature viruses were associated with a decreased dengue risk. Standard and mature serotype-specific nAb titers had the strongest protective effects against the matched serotype, supporting the dogma that exposure to one DENV serotype induces excellent immunity to that type[25]. However, when compared to the standard serotype-specific titers, the mature titers had bigger effect sizes against matched serotypes and decreased cross-reactivity among titers, suggesting that these viruses may be more selective in

detecting type-specific antibodies. Comparative PRNTs indicated that both maturation state and strain impact titer levels, but titers measured against the clinically mature strains had lower, more consistent thresholds associated with protection. Thus, in population and vaccine studies, the use of clinical and/or mature strains to measure titers may be preferable as correlates. In contrast, reference standard strains are more easily neutralized, potentially making them preferable for identifying any previous DENV exposure.

**Table 2 | Antibody measures associated with disease reduction as predicted by logistic regression adjusted for age, sex, and enrollment site**

| Disease Reduction | Case with any serotype | | | DENV2 disease | | DENV3 disease | |
|---|---|---|---|---|---|---|---|
| | ELISA IgG | Standard reference GMT | Mature clinical GMT | Standard reference DENV2 titer | Mature clinical DENV2 titer | Standard reference DENV3 titer | Mature clinical DENV3 titer |
| 50% | 2.6 | 87 | 42 | 79 | 20 | 72 | 23 |
| 70% | 4.5 | > 200 | 114 | > 200 | 40 | 181 | 40 |
| 90% | NA | NA | 344 | NA | 130 | NA | 140 |

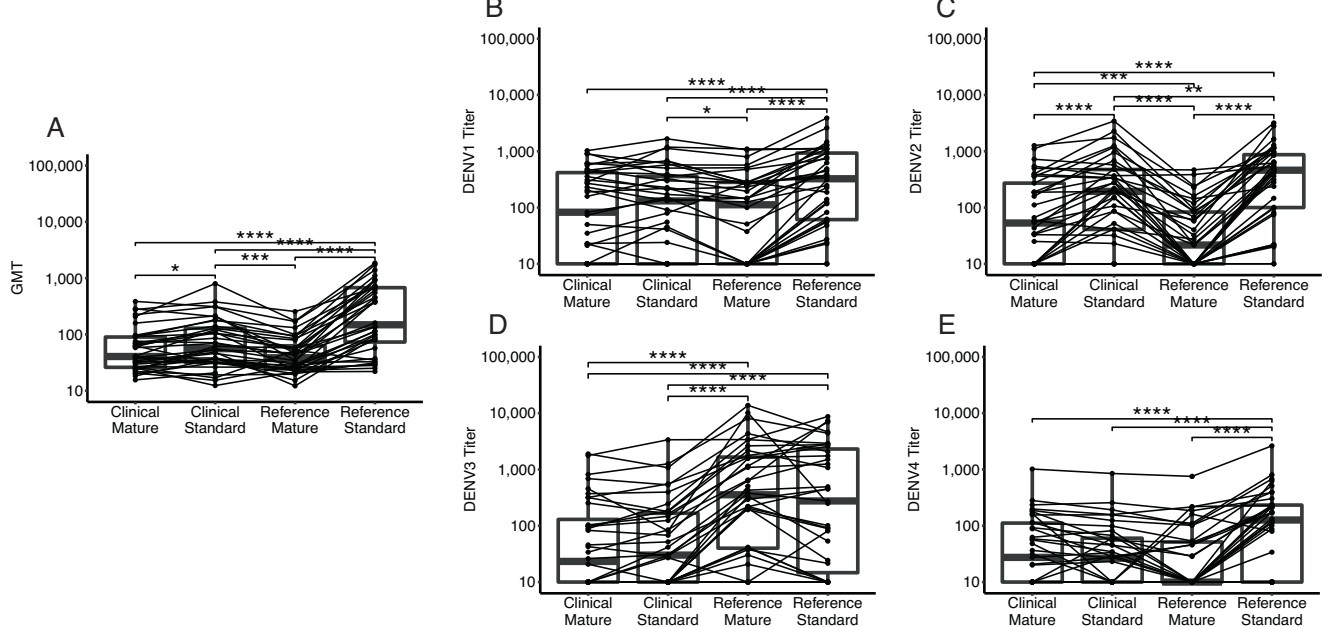

**Fig. 5 | Comparison of nAb titers by maturation state and virus strain.** Geometric mean (**A**) and serotype-specific (**B–E**) titers of each strain and maturation state combination. Horizontal lines connecting points track measurements from the same individual. Boxes indicate the interquartile range and bold lines indicate the median. $N = 36$, $*p < 0.05$, $**p < 0.01$, $***p < 0.001$, $****p < 0.0001$; two-sided paired $t$ test with Bonferroni correction. Source data are provided as a Source Data file.

We found that high total IgG and GMT measured by standard and mature viruses decreased the risk of dengue. We did not observe any enhancing effects at low to medium total binding antibodies and GMT. This may be because our cohort had few cases overall and almost no severe dengue cases. Previous work demonstrating enhancement at lower titers evaluated larger numbers of cases and identified the strongest enhancement signal for severe disease manifestations[26]. GMT, as measured with standard and mature viruses, was also associated with reduced odds of inapparent infection, which could be important for examining viral outbreak potential. The protective effects observed with high total binding and nAbs are consistent with findings from inhibition ELISA and vaccine-specific PRNT assays[13,27].

The type of virus used in the assay seems to impact the accuracy of the correlation. Specifically, although GMT > 200, as measured using standard reference strains, significantly decreased dengue risk, 27% of cases occurred among individuals with this antibody level. In addition, 43% of cases occurred among those with a standard GMT > 100. This is aligned with previous work where standard vaccine strains were used to measure GMT, and GMT > 100 at month 13 post-Dengvaxia was associated with 50% vaccine efficacy[13]. Conversely, antibodies, as measured with the mature clinical strains, had a lower threshold for protection, and only 2% of cases occurred among those with GMT > 100. Notably, 78% of our cohort had multitypic immunity, and GMT is likely a more accurate correlate in individuals with this immune history, as cross-reactive antibodies induced after secondary

infection may be providing protection. For monotypic individuals, the GMT is the average of a high titer against one serotype and low titers against the other three. Thus, the GMT would be accurate except against the strain that induced monotypic immunity. In sum, GMT is likely most useful as a correlate when measured with mature clinical strains in more multitypic populations.

Among all the antibodies and cases assessed, the mature clinical serotype-specific nAb titer against the matched serotype had the strongest effect sizes and lowest thresholds for protection. These protective effects at lower antibody levels suggest that mature virions bind more potent antibodies than standard virions. In addition, weaker correlations among the mature GMT and serotype-specific nAb suggest that mature clinical virions bind more type-specific antibodies than reference standard strains. Previous studies have demonstrated that quaternary type-specific and cross-reactive monoclonal antibodies neutralize both standard and mature viruses, while fusion loop antibodies only potently neutralize standard virus[22,28]. Ongoing and planned future studies will help determine when the mature neutralization assay is measuring type-specific versus cross-reactive antibodies, and whether the mature neutralization assay is as effective a correlate of protection as assays that measure these distinct antibody populations separately. Antibody depletion assays are the established method for identifying type-specific antibodies and have been associated with vaccine efficacy[29–31]. In a study of naïve CYD-TDV vaccinated individuals, 4/15 with breakthrough DENV1 cases and 4/18 with

breakthrough DENV3 had type-specific antibodies to the future infecting serotype[30]. In our study among those with natural primary or multitypic immunity at baseline, the number with mature neutralizing antibody titers to the future infecting serotype was similarly low, with 3/11 DENV2 cases and 4/18 DENV3 cases having pre-existing titers to the infecting serotype. In our study, it is possible that the mature neutralization assay is detecting broadly neutralizing antibodies against dengue, which cannot be detected by current depletion approaches but are believed to be important for long-term protection[32]. Ongoing studies to measure the presence and potency of known antibodies to the E dimer epitope in serum as correlates of protection and associations with mature neutralization assay titers will help inform the contribution of these cross-reactive antibodies to protection.

Interestingly, heterotypic titers were not associated with protection against DENV2 or DENV3, except that standard titers against DENV1 and DENV4 did decrease DENV3 and DENV2 disease risk, respectively. This protective effect and the tight correlation between the standard DENV1 and DENV3 titers suggest that these virions are likely binding to some of the same antibodies. Moreover, all serotype-specific standard nAb were strongly correlated with GMT ($r \geq 0.79$), while mature serotype-specific nAb had weaker correlations with GMT ($r < 0.79$). Tight associations between serotype-specific nAb and GMT are likely due to standard virions binding cross-reactive antibodies. In contrast, weaker correlations among clinically mature serotype-specific titers likely reflect more selective antibody binding. We hypothesize that some of this discrimination is due to the structure of mature virions with their less accessible fusion loop, but the use of clinical strains may also contribute. We also evaluated whether the magnitude of antibodies binding to certain antigens that are thought to be the targets of type-specific protection, including EDIII and NS1, were associated with dengue disease and found they were not strong predictors. This finding suggests that measuring a broader antibody pool may be important for dengue immune correlates.

Neutralization assays assessing the impact of strain and maturation state on titers revealed that the reference standard viruses had the highest titers, except against DENV3, where they were similar to the reference mature titers. Conversely, the reference mature virus induced the lowest titers against DENV2. Clinical standard and mature viruses induced similar titer levels, except against DENV2, where the clinical mature titers were lower. High titers against reference standard strains are consistent with previous work, indicating that highly laboratory-passaged DENVs are the most sensitive to neutralization[22]. The significant differences in titers measured among all DENV2 strains may reflect increased variability in epitope exposure and sensitivity to neutralization due to viral breathing[33]. DENV2 aside, the consistent titers among clinical strains suggest that these may bind primarily higher quality antibodies in either maturation state. Thus, the maturation state may have less influence on titer and virus neutralization when non-DENV2 clinical strains are used. Although these conclusions require confirmation with larger datasets examining more serotypes, this work suggests that clinical and mature strains are less sensitive to neutralization.

The decreased sensitivity to neutralization may result in the identification of higher-quality antibodies measured by mature and clinical strains, thereby lowering the threshold for detecting disease reduction. Specifically, while an ELISA value of 4.5 and a standard GMT > 200 were associated with 70% disease reduction, a mature GMT of 114 provided that level of protection. In addition, only titers measured with the mature clinical strains were associated with 90% disease reduction. However, reaching 90% disease reduction required very high titers suggesting that titers measured with the mature clinical strains may still be an unsatisfying correlate. Moreover, these are time intensive assays requiring specialized labor, which also may limit their utility for large studies.

Our study has several limitations. First, this cohort came from an observational study where guardians chose whether to vaccinate their children with CYD-TDV. Thus, this was non-randomized and may have limited generalizability. Specifically, we found that compared to the vaccinated children, the unvaccinated children were more likely to reside in Balamban vs. Bogo and had slightly lower baseline ELISA values. In addition, there may be other confounding variables that cannot be controlled for due to the observational design. Separately, given the difficulties of performing PRNTs, our data are limited by the use of random subsets and two dilution rather than full dilution assays for the standard reference strain. The two-dilution PRNT allowed us to estimate titers on more individuals, which has been validated in a separate cohort[34], and we found strong correlations between full titers and estimated two dilution titers when assessed with the mature clinical PRNT data (Supplementary Fig. 3). Inverse probability weighting was used to complete the datasets, and this was validated by comparing the estimates obtained using this method versus those from bootstrapping and from the full dataset using the ELISA measures (Supplementary Fig. 4). In addition, our cohort included no severe dengue cases, a high percentage of individuals with multitypic immunity, and infections mostly with DENV2 and DENV3. Thus, nAb thresholds associated with protection may differ in other populations and with the virus strain chosen for the assay. Inapparent infections were only evaluated in the first year using the mature viruses or Luminex assays because this time point was important for other planned analyses, and a limited number of total PRNT assays were performed. Moreover, inapparent infections may have been missed, especially in multitypic individuals, since their established immunity could prevent them from experiencing a significant rise in antibodies after exposure, as demonstrated in previous studies[35,36]. Despite their limitations, these data provide compelling support for the use of mature and clinical isolates in PRNTs and for titers obtained from these assays to serve as correlates of risk.

Overall, an accurate and reproducible correlate of protection would be a valuable contribution to the development of dengue vaccines. After adjusting for age, sex, and enrollment location, we demonstrate that ELISA IgG, GMT, and serotype-specific nAb titers measured by the standard and mature assays serve as correlates of risk for inapparent and symptomatic DENV infection. Mature and clinical strains likely identify higher quality nAbs given the virion structure and similarities with infecting strains, as evidenced by a lower, more consistent threshold associated with protection. Notably, immune correlates for naïve individuals who receive dengue vaccines are particularly important, given it is more difficult to protect this group. Our study focused on natural immunity, and future studies of the cohort are planned to evaluate these correlates in vaccinated individuals, although the numbers are small. Studies using mature neutralization assays in other cohorts and with other vaccines would be valuable for further evaluating the correlates described here, especially in a larger number of naïve individuals. In sum, nAb titers measured with mature and clinical isolates may be a helpful though exacting correlate of protection, and this should be further validated in vaccine studies with a focus on naïve populations.

## Methods

### Ethics statement

The study protocol was approved by the University of the Philippines – Manila Research Ethics Board and is registered at clinicaltrials.gov (NCT03465254). Written informed consent was obtained from a parent or legal guardian, and verbal assent was obtained from participants. The participants did receive compensation for their time, which included Php 500 during acute, convalescent, and annual blood collection. A subset of children also participated in the PBMC blood draw,

which required additional visits, and these received Php 1000. Travel costs for study visits were also reimbursed.

## Participants and baseline characteristics

Serum samples were collected from 2996 enrolled participants between May 2 and June 2, 2017, and febrile surveillance occurred between November 1, 2017 and October 31, 2022. All participants had basic demographic information collected. Sex was self-reported by the children or their guardians and was not considered in the design of the study since this was a convenience sample. Gender information was not collected because this was not included in the design of the protocol.

Febrile cases were evaluated with a case report form and reverse transcriptase polymerase chain reaction (RT-PCR) for DENV1-4[24]. Specifically, QIAmp viral RNA kits (QIAGEN, Valencia, CA, USA) were used to extract total nucleic acids from serum samples per the manufacturer's protocol. The Simplexa Dengue assay (Focus Diagnostics, Cypress, CA, USA) was then used to identify DENV infection and serotype[37]. This is a multiplex single-step real-time PCR assay. In short, reaction mixes for DENV1 & DENV4 and DENV2 & DENV3 were prepared per the manufacturer, which included serotype-specific primers, Taq polymerase, and RT enzyme. Then, 5μL of the mixes were added to Universal Disc wells (3M Focus Diagnostics) along with 5μL of the extracted RNA, Molecular Controls (MC, inactivated DENV1-4 serotypes), and a no template control (NTC). Wells were then sealed, inserted into the 3M Integrated Cycler instrument (3M Focus Diagnostics), and run per the manufacturer conditions. The Integrated Cycler Studio Software version 4.2 was used for data collection and analysis. Experiments were considered successful if there was positive detection of MC, negative detection of NTC, and negative samples had RNA IC amplification curves. Samples were considered positive for DENV infection if they had a Ct value $\leq 40$ and $\neq 0$ for any serotype.

There were 57 cases of symptomatic dengue among the unvaccinated (4.7%). Seven participants had a second dengue case during the observation period, but only the first case was considered here. Cases were classified as Dengue without Warning Signs (DwoWS), Dengue with Warning Signs (DWWS), and Severe Dengue. All baseline participant data were summarized using the gtsummary package. Means were compared using two-sided two-sample $t$ tests, and proportions were compared using Pearson's Chi-squared test.

## Sample selection

We designed a series of case-control studies to evaluate each serological assay as a correlate of dengue risk. We tested all available samples from individuals who progressed to symptomatic dengue in any immune group. Individuals who did not experience dengue during the study were included as controls. Given that many of these assays are labor intensive, subsets of the controls were tested by each immune assay, and these were chosen in the following manner. First, baseline anti-DENV immunity was evaluated in all individuals by indirect DENV IgG enzyme-linked immunosorbent assay (ELISA, Pan-Bio; Brisbane, QLD, Australia) according to the manufacturer's instructions. Since this ELISA uses inactivated DENV1-4 viruses as the antigen, it detects total antibodies[38]. Next, given the labor-intensive nature of the PRNT assay, testing using the standard reference strains was performed on a random selection of samples[23]. The standard PRNT was performed on 34% of those with ELISA < 0.2 and 30% of those with ELISA > 3, since these were expected to be naïve and multitypic, respectively (Supplementary Table 3). All individuals with available samples and an ELISA value of 0.2–3 had the standard PRNT performed to clarify their immune status. These subsets were used to calculate the weights for the models assessing the nAbs against the standard reference strains. Samples that neutralized at least 70% of infection at a serum dilution of 1:40 were considered immune to the tested serotype. Neutralization of one serotype was considered monotypic,

while neutralization of >1 serotype was labeled multitypic immunity. Samples that did not neutralize any serotype were considered naïve[23].

Subsets from these naïve, monotypic, and multitypic groups were then randomly selected for the assessment of nAb by mature PRNT, binding NS1 and EDIII antibodies, and the occurrence of inapparent infections (Supplementary Table 4). To account for the different proportions of samples tested within each baseline immune status group, we used inverse probability weighting in all models to ensure the sample was representative of the cohort. In addition, to have a consistent comparison group across models, we assumed undetectable mature PRNT and NS1 and EDIII antibodies in all individuals with ELISA < 0.2 ($n = 98$) or undetectable antibodies by standard PRNT ($n = 40$). Validation testing supported this assumption. Specifically, among individuals with ELISA < 0.2, 99% were naïve by standard PRNT ($n = 78$), and 100% were naïve by mature PRNT ($n = 36$).

## Standard and mature PRNT assays

To facilitate the analysis of many samples, the standard PRNT was simplified to consist of only two dilutions (1:40 and 1:200, hereafter reported as 40 and 200) performed in technical duplicates[23]. Antibody titers were measured using DENV1-4 WHO reference strains (DENV1 WP74, DENV2 S16803, DENV3 CH53489, DENV4 TVP-376) grown in C6/36 cells, and this assay is referred to as the standard reference PRNT (standard, where not otherwise indicated). $PRNT_{50}$ titers were estimated from the two-dilution assay in comparison to control wells with the virus but no serum using the Reed-Muench method[39].

To assess nAbs against mature virions, we used low-passage DENV strains that matched the genotypes circulating in the Philippines during the study period (GenBank accession numbers PQ667803, PQ667804, PQ667805, and PQ667806). We generated mature virus by propagating these viruses in Vero-furin cell lines, originally derived from Vero cells (ATCC CCL-81), and this assay is referred to as clinical mature (mature)[40,41]. The PRNT assay using standard or mature viruses was performed as follows: $1.7 \times 10^4$ Vero cells (ATCC CCL-81) per well were plated and incubated at 37 °C overnight. For the mature assays, human immune sera were fourfold serially diluted in technical duplicates over a range from 1:20 to 20480 and mixed with Vero-furin-produced low passage clinical isolates of DENV1-4. For both assays, sera-virus mixtures were incubated at 37 °C for one hour and then added to confluent cells for a one-hour inoculation. Subsequently, 1% methylcellulose or carboxymethyl cellulose overlay was added, and cells were incubated for 2 days. Cells were fixed in 80% methanol, blocked in 5% non-fat dried milk, and immunostained using mouse 4G2 and 2H2 anti-pan in flavivirus monoclonal antibodies diluted 1:2000, secondary horseradish peroxidase (HRP)-labeled goat anti-mouse IgG antibody diluted 1:3000 (KPL/SeraCare, catalog #: 5220-0341, lot #: 10506326), and developed using TrueBlue HRP substrate. Images of wells were collected using the Cellular Technology Limited (CTL) machine and ImmunoSpot software, and automated plaque counting was performed using the Viridot plaque counter package in R[42]. $PRNT_{50}$ titers were estimated by 4-parameter logistic regression at a 50% reduction in plaque count relative to virus-only control wells using the drc package in R.

For both the standard and mature PRNT, those with $PRNT_{50}$ titers outside the assay range were set as a two-fold dilution below or a two-fold dilution above the assay limit of detection. GMTs were calculated as the average of the log-transformed DENV1-4 nAb titers. To validate the two-dilution standard nAb titers, we reduced the full-dilution data from the mature PRNT to only 40 and 200 dilutions and estimated $PRNT_{50}$ titers using the Reed-Muench method. PRNT data from full dilution and two-dilution methods were highly correlated (Supplementary Fig. 3). Pearson correlations between full dilution and two-dilution mature serotype-specific titers and GMT were assessed using the ggpubr package and plotted with ggplot2. Only samples within the

limit of detection of the assay were included in these comparisons. This analysis was performed on multitypic individuals only since these are the most immunologically complex sera.

To assess the impact of strain and maturation state on nAb titers, a direct comparison of nAb titers against mature and standard WHO reference and circulating clinical strains was performed. Each reference and clinical strain was grown in Vero cells to produce standard isolates with prM proteins present and in furin-overexpressing Vero cells to produce fully mature viral particles. Six-dilution PRNT assays were performed as described above on a random subset of individuals with baseline multitypic ($n = 30$) or monotypic immunity ($n = 6$). The resulting GMT and serotype-specific titers against each standard and mature WHO and clinical strain were plotted using ggplot2 and ggpubr packages and compared using pairwise paired $t$-tests with a Bonferroni correction for multiple comparisons (rstatix package).

### Luminex multiplex assays
Antibody responses to envelope protein domain 3 (EDIII) and non-structural 1 protein (NS1) of DENV serotypes 1–4 were measured in a random subset of participants ($n = 298$, Supplementary Table 4) using Luminex multiplex assay[43]. Specifically, biotinylated EDIII antigens and biotinylated bovine serum albumin (BSA) were coupled to unique MagPlex®-Avidin Microspheres (Luminex), while His-tagged NS1 antigens (The Native Antigen Company) were coupled following immobilization of anti-His tag antibody (abcam) onto unique avidin-coated microspheres. The panel of EDIII, NS1, and BSA conjugated microspheres was mixed in equal ratios and plated at 2500 beads per antigen in 50 μL/well in 96 well plates. Diluted human serum (1:500) was incubated in singlicate with antigen-conjugated microspheres for one hour at 37 °C, 700 rpm. Later, immune complexes were incubated with 50 μL goat anti-human IgG Fc multi-species SP ads-PE antibody (Southern Biotech, catalog #: 2014-09) following three washes. Antibody responses were detected using a Luminex 200 analyzer and expressed as median fluorescence intensity after subtracting the non-specific antibody binding signal (to BSA). Selected samples from healthy donors and well-characterized DENV and ZIKV seropositive individuals were run on multiple assay plates to verify assay performance and assess inter-assay variability.

### Inapparent infections
The frequency of inapparent infection was assessed in a subset of participants (79 naïve, 68 monotypic, and 66 multitypic individuals). This was performed by comparing antibody levels at baseline versus those collected 1-2 years after enrollment. Antibody comparisons were made using the mature $PRNT_{50}$ titers for naïve and multitypic individuals and the Luminex results for the monotypic individuals. Consistent with previous work, seroconversion or a $\geq 4$-fold increase in mature $PRNT_{50}$ against any serotype without documented clinical symptoms was labeled an inapparent infection for multitypic and naïve individuals[44]. For monotypic individuals, $> 2$-fold rise in EDIII antibodies against 1 + serotype or NS1 antibodies against 3 + serotypes as measured by Luminex were defined as DENV infection and confirmed by mature PRNT[43].

### Correlates of risk analyses
All analyses were performed with RStudio for macOS (2022.07.1, Build 554). To account for potential non-linear relationships between antibody titers and outcomes, we estimated the probability of symptomatic dengue, dengue without warning signs, and dengue with warning signs as functions of baseline ELISA IgG and the geometric mean of standard and mature PRNT titers on both discrete and continuous scales using logistic regression, which assumes the log-odds (logit link function) of experiencing a case follows the binomial distribution. Inverse probability weighting was used to adjust for the number of sampled individuals within each immune group. Continuous

relationships were modeled using the mgcv package to create generalized additive models with 95% confidence intervals (CI) to allow for non-linear effects. The logistic regression models for binned antibody values were generated using the stats package. Antibody bins for ELISA data were chosen considering the manufacturer-recommended cut-point of $> 1.1$ as DENV seropositive and equalizing the numbers of samples in each bin as much as possible (Supplementary Table 5). For standard GMT data, antibody bins were chosen based on the two dilution points of 40 and 200 and an intermediate point of 100, which was noted by Gilbert et al. as the titer where dengue vaccine efficacy (Dengvaxia, Sanofi Pasteur) was consistently above 50%[13]. For mature GMT data, antibody bins were chosen based on the lowest dilution point of 20 and the intermediate point of 100. The probabilities from the generalized additive models and binned logistic regression models were then combined and graphed using the ggplot2 package. The predicted disease probabilities from each model were obtained using the binomial distribution and ggeffects package. Because DENV2 and DENV3 cases were less frequent events, the odds of each outcome were assessed by baseline standard and mature PRNT and serotype-specific EDIII and NS1 binding antibodies as continuous, linear predictors using the stats package and inverse probability weighting. Forest plots were generated using ggplot2. All models used logistic regression (assuming a binomial error distribution with a logit-link function) and were adjusted for age, sex, and enrollment site. Dengue probabilities are shown for the average study participant (age 10, female, from Bogo). All models were assessed for collinearity among predictors by using the car package to calculate the variance inflation factors, which were $< 5$ indicating no significant collinearity.

To evaluate how models may have been affected by inverse probability weighting, we performed a sensitivity analysis where we limited the ELISA dataset to values only from those with binding antibody (Luminex, $n = 298$), standard ($n = 823$), or mature ($n = 293$) nAb titer data. We then built models of each of these datasets adjusted by inverse probability weighting and bootstrapping and compared the results to those gained from the model of the full ELISA dataset. Bootstrapping was performed using the boot package in R, wherein subsets were resampled 2000 times, and CIs were calculated using bias-corrected and accelerated bootstraps. CIs from the full dataset largely overlapped with the models from weighted and bootstrapped datasets (Supplementary Fig. 4). Thus, the effect sizes from the models using weighted binding and nAb as predictors likely reflect the true effect sizes from the full dataset.

### Reporting summary
Further information on research design is available in the Nature Portfolio Reporting Summary linked to this article.

## Data availability
The source data generated in this study have been deposited in the Zenodo database under accession code (https://doi.org/10.5281/zenodo.13941973). Informed consent forms are available in the supplementary materials. This study did not generate new unique reagents. Further information and requests for resources and reagents should be directed to and will be fulfilled by the lead contact, Leah C. Katzelnick (leah.katzelnick@nih.gov). Source data are provided in this paper.

## Code availability
The code has been deposited at Zenodo and is freely accessible (https://doi.org/10.5281/zenodo.13941973).

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

## Acknowledgements
We thank all the study participants, our local collaborators, and field staff. We also thank V. Tse and R. Baric for providing the Vero-furin overexpressing cell line. This work was supported in part by the Division of Intramural Research at the US National Institute of Allergy and Infectious Diseases (NIAID) to LCK. The cohort was funded by the Philippine Department of Health, Hanako Foundation, World Health Organization, Swedish International Development Cooperation Agency through the International Vaccine Institute, and University of North Carolina at Chapel Hill, and US NIAID grant # P01AI106695 to J.D. Numerous serological analyses of these samples were supported by # P01AI106695 to AMDS.

## Author contributions
M.Y. and J.D. conceived the cohort study and wrote the study protocol. J.D., M.Y., J.V.D., M.V.C. and K.A.A., supervised, implemented, and guided the cohort study procedures. M.Y., J.V.D., M.V.C., J.D. and K.A.A. implemented field activities. M.Y., M.V.C., J.V.D. and K.A.A. trained and monitored the staff on data collection and management. J.V.D. trained the staff on sample collection, storage, and transport, and performed the dengue enzyme-linked immunosorbent assay. C.A., L.J.W. and A.M.D.S. carried out the standard plaque reduction neutralization testing on baseline samples to establish an immune profile. A.C.E., L.C.K., L.J.W., P.M., G.R.R., S.F., C.V. and A.M.D.S. carried out the paired immunogenicity testing using the mature plaque reduction neutralization test of baseline and follow-up samples. L.D.H., A.M.D.S. and M.A.F. designed and performed the Luminex assays. C.D.O. and L.C.K. conceived the immune correlates study and designed the antibody analyses and models. C.D.O. performed all immune correlate analyses. R.A.A. informed and supported model design and analysis. C.V. performed analyses comparing titers by strain and maturation state. C.D.O. and L.C.K. drafted the manuscript, and all authors contributed to the revision of the manuscript.

## Competing interests
M.Y., M.V.C., J.V.D., K.A.A., A.M.C. and A.K.S. report receiving salaries from 2017 onwards as part of an ongoing separate study (effectiveness of the tetravalent dengue vaccine, CYD-TDV [Dengvaxia] in the Philippines) sponsored by the University of the Philippines Manila and funded by Sanofi Pasteur. J.D. was an unpaid external consultant in the Extended Study Group for dengue vaccine effectiveness evaluation studies in Asia in 2015 convened by Sanofi Pasteur and is an unpaid investigator of an ongoing separate study (effectiveness of the tetravalent dengue vaccine, CYD-TDV [Dengvaxia] in the Philippines) sponsored by the University of the Philippines Manila and funded by Sanofi Pasteur. The protocol was written to evaluate the effect of the Sanofi Pasteur Dengvaxia vaccine. However, only unvaccinated individuals were included in the present manuscript. All other authors declare no competing interests.

## Ethics
J.D., M.Y., J.V.D., M.V.C., and K.A.A. are investigators local to the Philippines who designed and established the cohort and collected all the demographic, clinical, and blood samples, and generated the enzyme-linked immunosorbent assay data. All local investigators were included in discussions of this work evaluating antibodies as correlates, roles, and responsibilities were agreed upon ahead of the research, and authorship was determined accordingly. This research is highly relevant to Cebu, Philippines, given the heavy local burden of disease, and this work was funded, in large part, by the Philippine Department of Health. Capacity building included additional training for JVD in the laboratory AMDS. This research did not result in stigmatization, incrimination, discrimination, or otherwise personal risk to participants. Related work by the local investigators has been cited appropriately here[23,24].
