## [Peer Review file · Nature Communications]

Dengue virus IgG and neutralizing antibody titers measured with standard and mature viruses are protective

Corresponding Author: Dr Leah Katzelnick

Version 0:

Reviewer comments:

Reviewer #1

(Remarks to the Author)

The manuscript describes statistical modeling of neutralizing antibody titers as a potential correlate of risk in the study of Dengue disease. Of particular interest is the selection of viruses for the assay (reference/clinical, standard/mature), where mature viruses are generally less sensitive.

My biggest concern is in the methodology. Line 243: describing the regression model "... continuous scales using logistic regression in the Poisson distribution". The Poisson model is typically used for (and often synonymous with) a regression of count data with the log link function. In fact, the log link is the default in R when the Poisson family is specified (see ?family).

1. I looked at the code (10.5281/zenodo.10892335) and I did not see a specification of the logit link (i.e., family = Poisson(link = "logit")), so a log regression seemed to be used, not a logistic regression (there is one case where family = binomial, which would use the logit link, was this intended?). Assuming this was the code used, the exponentiated coefficients of the log regressions that are presented in this study are risk ratios, not odds ratios.
2. Can the authors provide justification for the use of the Poisson error model vs. binomial family for binary data? The following reference could provide that justification: <https://doi.org/10.1093/aje/kwr183> and perhaps there are extensions to GAMs.
3. If the authors determine their current model is suitable (including whether the standard errors are correctly specified), they will need to update the ratio interpretations accordingly.

The authors are analyzing a subset of participants from a larger observational study. The selection criteria for inclusion: "...only unvaccinated children were included in these analyses." (lines 133-134).

1. This does not appear to be a randomized subset, so the analysis cohort is determined by self(guardian)-selection to be vaccinated. The authors should comment on this selection bias more directly and how it limits the generalizability of the results, particularly in the discussion.
2. If they have access to the vaccinated cohort data, it would be interesting to see how the cohorts differ via baseline covariates. I.e., how comparable are the two groups to make a more general inference.

I was particularly interested in which experimental virus generates the highest quality correlate. The authors allude to this via effect size (ie, the magnitude of the regression coefficients). It is not easy to see this comparison in the current visualizations and how consistent it is across subtypes. Are there other metrics that could help illustrate correlate performance like an ROC?

(Minor) Two ELISA cutoffs are described Supp Table 1 and Supp Table 3. The latter is described in detail and used in the main results. What is the rationale for the cutoffs used in supp table 1?

Reviewer #2

(Remarks to the Author)

This study attempts to advance understanding of immune correlates of protection for dengue infection and disease, an area of vaccine immunology with many lacunae, i.e., a definitive correlate of protection has not been established for dengue vaccines. The authors compare baseline dengue IgG ELISA and plaque reduction neutralizing assays (PRNTs) using WHO reference standard viruses (containing a mix of mature and immature viruses) and mature dengue viruses using standard

and circulating Philippine dengue virus strains, as well as envelope domain III (EDIII) and non-structural protein 1 (NS1) binding antibodies with a Luminex multiplex assay for their relative protection against dengue disease in a subset cohort of 1206 unvaccinated Philippine children eligible for CYD-TDV dengue vaccine in a 5 year longitudinal study from 2017 - 2022. Two dilution PRNTs at 1:40 and 1:200 were used after validating with full dilution mature virus data. GMTs and serotype specific titers were measured. 57 cases of symptomatic dengue were identified, none with severe disease, but 32 had warning signs using the WHO case definitions. Varying case-control studies using different statistically adjusted subsamples of dengue naïve, dengue monotypic immune, and multitypic immune children were done to make the comparisons. Both high ELISA and PRNT antibody titers were associated with dengue disease protection, while EDIII and NS1 binding antibodies did not provide such protection. Mature virus PRNTs and PRNTs using clinical strains protected at lower titers and with more type specificity. The authors conclude that the different assays may be useful for different purposes, e.g. ELISA antibodies or standard reference PRNTS to determine prior dengue exposure, and mature virus and clinical isolate assays merit further investigation as to their potential utility as a "more exacting" correlate of protection for vaccine studies.

This well written and well designed study rigorously assesses relevant antibody assays for their protection against dengue disease. The authors appropriately describe study limitations, the limited number of dengue cases including no severe cases, mostly dengue virus 2 and 3 serotypes, and relatively few dengue naives among the children in the cohort.

Specific comments for authors

1) Methods, line 208, Figure 3, and Discussion – How confident are the authors that measuring 4-fold rises in antibody titers in previously dengue exposed children at one year would detect all asymptomatic infections. How many asymptomatic infections were detected? Could some children, particularly multitypics, have had antibody rises and falls back to baseline levels? Why was the asymptomatic infection rate measured only after the first year?

2) Discussion – The dengue vaccines licensed to date, CYD-TDV and TAK-003, have good efficacy in persons previously exposed to dengue. The most important issue with regard to the question of immune correlates of protection is thus identifying a useful biomarker in dengue naives. This point could be addressed in the discussion, even though the study included relatively few dengue naives.

3) Discussion – As mentioned in the introduction (lines 88-90), considerable evidence suggests that homotypic antibody identified with depletion assays provides protection against dengue. Would it not have been useful to perform depletion assays in cases with multitypic antibody in the study? Both depletion assays and mature virus PRNT assays are labor intensive, as mentioned. Perhaps the authors could comment on the next research steps to identify as close to ideal correlate of protection biomarker(s) for dengue measuring homotypic and broadly cross protecting antibody.

Editorial

4) Abstract, line 30-31 – Suggest that abstract specify what level of titer is meant by "high geometric mean PRNT titers" for both standard and mature virus assays.

5) Introduction, line 57 – Suggest "...but this determination can be complicated..."

6) Introduction, Line 59 – Suggest "... at one time point, threshold antibody levels may not persist due to waning immune responses."

7) Results, line 318 – Suggest "...had high mature PRNT > 100."

8) Figure 5 is a little hard to read. Could the boxes with interquartile range and median be made bolder.

9) References 13 and 24 lack the journal edition number, etc.

Version 1:

Reviewer comments:

Reviewer #1

(Remarks to the Author)

I want to thank the authors for thoroughly addressing all of my comments. I don't have any further questions or major concerns, nice work.

Reviewer #2

(Remarks to the Author)

The authors have thoroughly and satisfactorily addressed this reviewer's previous comments in the revised manuscript. As mentioned previously, the manuscript rigorously assesses the relevant antibody assays for the degree of protection against dengue disease, in the context of dengue virus serotypes 2 and 3 circulation and a limited number of children previously unexposed to dengue.

I do still have a question about the asymptomatic infection determinations in the subset of the children dengue naïve, with monotypic antibody and multitypic antibody. Why do the numbers tested change between the first submission and the revised manuscript (48N, 42 Mono in the first and 79N, 68 Mono in the revision)? According to Figure 3 91 or 92 asymptomatic infections were identified out of the subset of 213 individuals in the first year. This overall asymptomatic infection rate of 43% in one year is extremely high and seems inconsistent with 57 symptomatic infections ascertained after 5 years, a 4% illness incidence rate in the cohort of unvaccinated children. Do the authors think these 92 children have been aviremically boosted?

Also line 157 in Methods indicates testing for asymptomatic infection was done "approximately one year after enrollment."

Line 357 in Results indicates infections ascertained "over the first 1-2 years of follow-up).

Minor editorial

1) Line 489 - Replace "it" with "is."

2) Suggest replacing reference 7 on TAK-003 dengue vaccine trial with the final trial publication by Tricou et al, Lancet Global Health 12(2) E257-270, Feb 2024

Responses to Reviewers:

Reviewer 1: The manuscript describes statistical modeling of neutralizing antibody titers as a potential correlate of risk in the study of Dengue disease. Of particular interest is the selection of viruses for the assay (reference/clinical, standard/mature), where mature viruses are generally less sensitive.

My biggest concern is in the methodology. Line 243: describing the regression model "... continuous scales using logistic regression in the Poisson distribution". The Poisson model is typically used for (and often synonymous with) a regression of count data with the log link function. In fact, the log link is the default in R when the Poisson family is specified (see ?family).

- 1. I looked at the code (10.5281/zenodo.10892335) and I did not see a specification of the logit link (i.e., family = Poisson(link = "logit")), so a log regression seemed to be used, not a logistic regression (there is one case where family = binomial, which would use the logit link, was this intended?). Assuming this was the code used, the exponentiated coefficients of the log regressions that are presented in this study are risk ratios, not odds ratios.**
- 2. Can the authors provide justification for the use of the Poisson error model vs. binomial family for binary data? The following reference could provide that justification: <https://doi.org/10.1093/aje/kwr183> and perhaps there are extensions to GAMs.**
- 3. If the authors determine their current model is suitable (including whether the standard errors are correctly specified), they will need to update the ratio interpretations accordingly.**

A: We thank the reviewer for this excellent point. In our cohort, the samples were selected for evaluation based on outcome. Specifically, all individuals who developed dengue had their nAbs measured, a subset of those who did not develop dengue served as controls, and weighting was used to adjust for the subset size. With this design, odds ratios are preferable to risk ratios, which are favored when the samples are selected based on the risk factors rather than outcomes. To obtain odds ratios, we had originally used binomial models with link = "logit." The models were switched in error at a late stage of analysis and have now been reverted back to binomial(link = "logit") in the code. This change had no effect on significance of our results. Indeed, the effect sizes are now generally slightly larger than when using the Poisson models. We checked the manuscript to ensure all odds ratios are reported correctly and the interpretation of the odds ratios are correctly reported. We have changed the methods as follows:

Methods: line 257 in the clean version of the manuscript

"To account for potential non-linear relationships between antibody titers and outcomes, we estimated the probability of symptomatic dengue, dengue without warning signs, and dengue with warning signs as functions of baseline ELISA IgG and the geometric mean of standard and mature PRNT titers on both discrete and continuous scales using logistic regression, which assumes the log-odds (logit link function) of experiencing a case follows the binomial distribution."

Methods, line 280:

“All models used logistic regression (assuming a binomial error distribution with a logit-link function) and were adjusted for age, sex, and enrollment site. Dengue probabilities are shown for the average study participant (age 10, female, from Bogo).”

The authors are analyzing a subset of participants from a larger observational study. The selection criteria for inclusion: "...only unvaccinated children were included in these analyses." (lines 133-134).

1. This does not appear to be a randomized subset, so the analysis cohort is determined by self(guardian)-selection to be vaccinated. The authors should comment on this selection bias more directly and how it limits the generalizability of the results, particularly in the discussion.

A: We completely agree with the reviewer. We have stated this more explicitly in the methods section and described the associated limitations in the discussion section as follows:

Methods, line 135:

“In total, n=1790 guardians elected for their children to receive a single dose of the vaccine while n=1206 chose for their children not to be vaccinated.”

Discussion, line 522:

“First, this cohort came from an observational study where guardians chose whether to vaccinate their children with CYD-TDV. Thus, this was non-randomized and may have limited generalizability. Specifically, we found that compared to the vaccinated children, the unvaccinated children were more likely to reside in Balamban vs. Bogo and had slightly lower baseline ELISA values. Additionally, there may be other confounding variables that cannot be controlled for due to the observational design.”

2. If they have access to the vaccinated cohort data, it would be interesting to see how the cohorts differ via baseline covariates. I.e., how comparable are the two groups to make a more general inference.

A: We agree this is useful information. We have added the following analysis to the Methods and included a new supplemental table as follows:

Methods, line 137:

“The unvaccinated and vaccinated children had similar age, sex, and baseline serostatus (Supplemental Table 1). However, the unvaccinated group was more likely to reside in Balamban (51% vs. 46%, p=0.004) and had lower baseline mean dengue IgG ELISA (p=0.02).”

I was particularly interested in which experimental virus generates the highest quality correlate. The authors allude to this via effect size (ie, the magnitude of the regression coefficients). It is not easy to see this comparison in the current visualizations and how

consistent it is across subtypes. Are there other metrics that could help illustrate correlate performance like an ROC?

A: We agree that ROC curves can be very useful to illustrate performance, especially for a diagnostic test. However, it is difficult to interpret ROC curves for a predictive marker like an immune correlate, as among those who don't experience cases, there are those who would have experienced a case but simply were not yet exposed. This makes interpretation of sensitivity and specificity difficult. Instead, to compare the protective effects between antibody measures, we estimated the antibody level at which specific quantities of disease reduction were observed. We realize that our analysis may not have been sufficiently clearly described in the manuscript and have updated the results and discussion sections as follows:

Results, line 392:

“To compare the quality of the correlates, we assessed the antibody thresholds associated with specific percent reductions in dengue disease compared to being naive (Table 2), similar to reporting vaccine efficacy associated with specific antibody titer quantities¹⁴.”

Results, line 400:

“Thus, serotype-matched mature titers had the strongest associations with protection at the lowest levels.”

Discussion, line 460:

“Among all the antibodies and cases assessed, the mature clinical serotype-specific nAb titer against the matched serotype had the strongest effect sizes and lowest thresholds for protection. These protective effects at lower antibody levels suggests that mature virions bind more potent antibodies than standard virions.”

Additionally, we have added a visualization of the raw serotype-matched titers for all cases and non-cases in the study, now Figure 4 C and D, reproduced below. These results are described in the results:

Results, line 371:

“When visualizing the titer data directly, there was a difference in titer magnitude between cases and non-cases by both assays, but most of those with breakthrough cases had mature serotype matched titers below the assay limit of detection of 1:20, whereas a larger fraction of breakthrough cases had standard serotype matched titers above the assay limit of detection of 1:40 (Figure 4 C and D).”

Figure 4. Comparison of dengue caused by DENV2 (diamond) and DENV3 (inverted triangle) to non-cases by baseline GMT and serotype-specific nAbs. The odd ratios and 95% confidence intervals of experiencing a case of DENV2 or DENV3 by baseline titer measured with standard reference (A) or mature clinical (B) strains. All models were adjusted for age, sex, and enrollment site, and inverse probability weighting was used to adjust for GMT and nAb subset sizes. Individual-level data and distributions (medians with interquartile ranges) of baseline titers for primary and multitypic immune participants who experienced a case caused by each serotype or not, stratified by standard reference (C) or mature clinical (D) neutralization assay results.

(Minor) Two ELISA cutoffs are described Supp Table 1 and Supp Table 3. The latter is described in detail and used in the main results. What is the rationale for the cutoffs used in supp table 1?

A: Thank you for this question. Supplemental table 1 (now supplemental table 2) reflect the cut-offs originally used for selecting samples to test by PRNT assay using the standard reference strains, as described in Lopez et al. 2020, Lancet Global Health. These numbers were then used to calculate the weights for the models assessing the nAbs against the standard reference strains. This explanation was added to the methods section as follows:

Methods, line 157:

“Next, given the labor-intensive nature of the PRNT assay, testing using the standard reference strains was performed on a random selection of samples, as described previously²². The standard PRNT was performed on 34% of those with ELISA<0.2 and 30% of those with ELISA>3, since these were expected to be naïve and multitypic respectively (Supplemental Table 2). All individuals with an ELISA value of 0.2-3 had the standard PRNT performed to clarify their immune status. These subsets were used to calculate the weights for the models assessing the nAbs against the standard reference strains.”

Reviewer #1 (Remarks on code availability):

The code relies on data that I do not have access to (or at least I don't think I do), so I could not run it.

A: We have corresponded with our colleagues leading the Philippines cohort and they have agreed to the following code and data availability statement. This has been added to the manuscript (line 567-573):

Code and data availability: The code has been deposited at Zenodo and is freely accessible (10.5281/zenodo.10892334). De-identified participant epidemiological, clinical, and laboratory data, and informed consent forms can be made available according to the University of the Philippines Manila, the University of North Carolina, and the US National Institutes of Health data sharing policy on request to Dr. Jacqueline Deen (deen.jacqueline@gmail.com) and Dr. Michelle Ylade (mcylade@up.edu.ph), starting from the time of publication and for the subsequent 3 years.

I skimmed the code briefly to answer a question I had about the methods. There were two files: a script with a set of functions and then an rmarkdown to produce the analysis. It was generally accessible as a reference to someone with R expertise. However, there was limited documentation.

A: Thank you for this point. The R code has been edited to expand the documentation including adding explanatory headers to the functions and expanding the titles of each chunk in the Rmarkdown. When appropriate and to facilitate evaluation of the code, these titles now include the corresponding figure or table generated by the chunk.

Reviewer #2 (Remarks to the Author):

This study attempts to advance understanding of immune correlates of protection for dengue infection and disease, an area of vaccine immunology with many lacunae, i.e., a definitive correlate of protection has not been established for dengue vaccines. The authors compare baseline dengue IgG ELISA and plaque reduction neutralizing assays (PRNTs) using WHO reference standard viruses (containing a mix of mature and immature viruses) and mature dengue viruses using standard and circulating Philippine dengue virus strains, as well as envelope domain III (EDIII) and non-structural protein 1 (NS1) binding antibodies with a Luminex multiplex assay for their relative protection against dengue disease in a subset cohort of 1206 unvaccinated Philippine children eligible for CYD-TDV dengue vaccine in a 5 year longitudinal study from 2017 - 2022. Two dilution PRNTs at 1:40 and 1:200 were used after validating with full dilution mature virus data. GMTs and serotype specific titers were measured. 57 cases of symptomatic dengue were identified, none with severe disease, but 32 had warning signs using the WHO case definitions. Varying case-control studies using different statistically adjusted subsamples of dengue naïve, dengue monotypic immune, and multitypic immune children were done to make the comparisons. Both high ELISA and PRNT antibody titers were associated with dengue disease protection, while EDIII and NS1 binding antibodies did not provide such protection. Mature virus PRNTs and PRNTs using clinical strains protected at lower titers and with more type specificity. The authors conclude that the different assays may be useful for different purposes, e.g. ELISA antibodies or standard reference PRNTs to determine prior dengue exposure, and mature virus and clinical isolate assays merit further investigation as to their potential utility as a “more exacting” correlate of protection for vaccine studies.

This well written and well designed study rigorously assesses relevant antibody assays for their protection against dengue disease. The authors appropriately describe study limitations, the limited number of dengue cases including no severe cases, mostly dengue virus 2 and 3 serotypes, and relatively few dengue naives among the children in the cohort.

A: We thank the reviewer for this positive feedback.

Specific comments for authors

1) Methods, line 208, Figure 3, and Discussion – How confident are the authors that measuring 4-fold rises in antibody titers in previously dengue exposed children at one year would detect all asymptomatic infections.

A: This is an excellent point. Although this is a limited and imperfect measure of inapparent infections, a ≥ 4 -fold rise in neutralization titers is a standard in the dengue field (e.g., Montoya et al. 2013, PLOS NTD), although it was originally developed to identify infections due to a rise between acute and early convalescent samples, as recommended by the CDC. A ≥ 4 -fold rise has commonly been applied to identifying infections in those with pre-existing antibodies using neutralization titers, hemagglutination inhibition assay titers, and iELISA data, but is known to miss even symptomatic infections (Gordon et al. 2013 PLOS NTD), and other predictive models identify 10% more infections (Hamins-Puertolas et al. 2024 Nature Microbiology).

Additionally, due to the aims of other projects planned with this cohort, all monotypic individuals were assessed for inapparent infections using the Luminex data, with validation of a subset of the samples by mature neutralization assay. The schema used to determine inapparent infections with this assay is briefly described here and validated in a separate paper currently in revision at the Lancet Microbe (Hein Dahora, et al. 2024).

To clarify these points, we modified the methods and the discussion as follows:

Methods, line 246:

“The frequency of inapparent infection was assessed in a subset of participants (79 naïve, 68 monotypic, and 66 multitypic individuals). This was performed by comparing antibody levels at baseline versus those collected approximately 1 year after enrollment. Antibody comparisons were made using the mature PRNT₅₀ titers for naïve and multitypic individuals and the Luminex results for the monotypic individuals. Consistent with previous work, seroconversion or a ≥ 4 -fold increase in mature PRNT₅₀ against any serotype without documented clinical symptoms was labeled an inapparent infection for multitypic and naïve individuals³³. For monotypic individuals, >2 -fold rise in EDIII antibodies against 1+ serotype or NS1 antibodies against 3+ serotypes as measured by Luminex were defined as DENV infection and confirmed by mature PRNT³².”

Discussion, line 541:

“Inapparent infections may have been missed, especially in multitypic individuals, since their established immunity could prevent them from experiencing a significant rise in antibodies after an exposure, as demonstrated in previous studies^{44, 45}.”

How many asymptomatic infections were detected? Could some children, particularly multitypics, have had antibody rises and falls back to baseline levels?

A: These are important questions, thank you. We have reported the number of individuals with asymptomatic infections by baseline serostatus in the results:

Results, lines 345:

“Notably, among individuals who did not have symptomatic dengue, inapparent infections occurred in 54% of naïve, 49% of monotypic, and 35% of multitypic individuals.”

Why was the asymptomatic infection rate measured only after the first year?

A: The primary focus of our paper was on correlates of protection against symptomatic infection, which is why we focus on prevention of dengue cases. We included an analysis of inapparent infections as a comparison to prevention of symptomatic infections. Inapparent infections were only measured in the first year because this time point was important for other planned analyses and a thus PRNT assays were performed. There is ongoing work to measure inapparent infections for all five years of the cohort, but this is a major undertaking for the team and will not be completed for multiple years. We agree that this is a limitation and have added this to the discussion section as follows:

Discussion, line 538:

“Inapparent infections were only evaluated in the first year using the mature viruses or Luminex assays because this time point was important for other planned analyses and a limited number of total PRNT assays were performed.”

2) Discussion – The dengue vaccines licensed to date, CYD-TDV and TAK-003, have good efficacy in persons previously exposed to dengue. The most important issue with regard to the question of immune correlates of protection is thus identifying a useful biomarker in dengue naives. This point could be addressed in the discussion, even though the study included relatively few dengue naives.

A. This is an excellent point, and we have addressed it in the discussion as follows:

Discussion, line 552:

“Notably, immune correlates for naïve individuals who receive dengue vaccines are particularly important given it is more difficult to protect this group. Our study focused on natural immunity, and future studies of the cohort are planned to evaluate these correlates in vaccinated individuals, although numbers are small. Studies using mature neutralization assays in other cohorts and with other vaccines would be valuable for further evaluating the correlates described here, especially in a larger number of naïve individuals.”

3) Discussion – As mentioned in the introduction (lines 88-90), considerable evidence suggests that homotypic antibody identified with depletion assays provides protection against dengue. Would it not have been useful to perform depletion assays in cases with multitypic antibody in the study? Both depletion assays and mature virus PRNT assays are labor intensive, as mentioned. Perhaps the authors could comment on the next research steps to identify as close to ideal correlate of protection biomarker(s) for dengue measuring homotypic and broadly cross protecting antibody.

A: We agree that this is an important point, and one that we reference in the introduction. We have revised the introduction to make clear that we believe a strength of the mature neutralization assay is that it generically measures protective antibodies, without the need to distinguish type-specific and cross-reactive. Line 93 now reads:

“Assays that use mature rather than standard viruses to measure nAbs may better distinguish antibody quality and serve as more accurate correlates of protection, measuring both protective type-specific and cross-reactive antibodies^{19, 20}.”

We also revised our discussion to comment on our results and future next steps for identifying ideal biomarkers for measuring homotypic and broadly protective antibodies.

Discussion, line 460

“Among all the antibodies and cases assessed, the mature clinical serotype-specific nAb titer against the matched serotype had the strongest effect sizes and lowest thresholds for protection. These protective effects at lower antibody levels suggests that mature virions bind more potent antibodies than standard virions. Additionally, weaker correlations among the mature GMT and serotype-specific nAb suggest that mature clinical virions bind more type-specific antibodies than reference standard strains. Previous studies have demonstrated that quaternary type-specific and cross-reactive monoclonal antibodies neutralize both standard and mature viruses, while fusion loop antibodies only potently neutralize standard virus^{22,37}. Ongoing and planned future studies will help determine when the mature neutralization assay is measuring type-specific versus cross-reactive antibodies, and whether the mature neutralization assay it as effective a correlate of protection as assays that measure these distinct antibody populations separately. Antibody depletion assays are the established method for identifying type-specific antibodies and have been associated with vaccine efficacy^{38,39,40}. In a study of naïve CYD-TDV naïve vaccinated individuals, 4/15 with breakthrough DENV1 cases and 4/18 with breakthrough DENV3 had type-specific antibody to the future infecting serotype³⁹. In our study among those with natural primary or multitypic immunity at baseline, the number with mature neutralizing antibody titers to the future infecting serotype was similarly low, with 3/11 DENV2 cases and 4/18 DENV3 cases having pre-existing titers to the infecting serotype. In our study, it is possible that the mature neutralization assay is detecting broadly neutralizing antibodies against dengue, which cannot be detected by current depletion approaches but are believed to be important for long-term protection⁴¹. Ongoing studies to measure the presence and potency of known antibodies to the E dimer epitope in serum as correlates of protection and association with mature neutralization assay titers will help inform the contribution of these cross-reactive antibodies to protection.”

With respect to our planned studies, our co-author Dr. Aravinda de Silva is working to directly compare type-specific antibodies measured by depletion assays and mature neutralizing antibody titers in vaccinated and unvaccinated subjects and across immune groups to measure type-specific antibodies as correlates of protection in this cohort. Additionally, since the submission of this manuscript, we have submitted another manuscript describing the role of cross-reactive antibodies targeting the E dimer epitope in explaining protection among multitypic children in the same cohort study in the Philippines. The preprint can be found here: <https://doi.org/10.1101/2024.04.30.24306574>.

Editorial

4) Abstract, line 30-31 – Suggest that abstract specify what level of titer is meant by “high geometric mean PRNT titers” for both standard and mature virus assays.

A: Thank you for this suggestion. We have edited the abstract to read:

“Standard PRNT geometric mean titers (GMT) >200 and mature GMT >100 were associated with reduced dengue disease overall (p<0.01)...”

5) Introduction, line 57 – Suggest “...but this determination can be complicated...”

A: Thank you for this suggestion. This sentence (now line 55) has been edited as follows:

“Once identified, regulators prefer that a correlate of protection have a single threshold that distinguishes those who get disease from those who do not get disease or even infection, but this determination can be complicated by heterogeneity in exposures and responses.¹¹”

6) Introduction, Line 59 – Suggest “... at one time point, threshold antibody levels may not persist due to waning immune responses.”

A: Thank you for this suggestion. This sentence (now line 58) has been edited as follows:

“Additionally, although an intervention meets the correlate of protection threshold at one time point, threshold antibody levels may not persist due to waning immune responses.”

7) Results, line 318 – Suggest “...had high mature PRNT > 100.

A: Thank you for this suggestion. This sentence (now line 336) has been edited as follows:

“Notably, 29% of those with standard GMT>200 had mature GMT>100, and only 1% of those with standard GMT<200 had mature GMT>100.”

8) Figure 5 is a little hard to read. Could the boxes with interquartile range and median be made bolder.

A: We agree with the reviewer and have darkened the boxes and interquartile ranges.

9) References 13 and 24 lack the journal edition number, etc.

A: Thank you for these points. These references have been updated.

REVIEWER COMMENTS

Reviewer #1 (Remarks to the Author):

I want to thank the authors for thoroughly addressing all of my comments. I don't have any further questions or major concerns, nice work.

A: We thank the reviewer for their thoughtful edits of our work. Addressing the comments has greatly improved our manuscript.

Reviewer #1 (Remarks on code availability):

The code provided via this reference I traced to link <https://zenodo.org/records/10892335>. This was the code of the original submission. In the rebuttal, the authors provided 10.5281/zenodo.10892334, but I was unable to trace this reference to any link. Given the author responses, only minor edits to the original submission were required to address my main comments besides providing the primary data source, which was also addressed in the rebuttal.

A: Thank you for pointing this out. As mentioned in the prior review, we did update the code to reflect the binomial distribution and expanded the documentation including adding explanatory headers referencing the specific figures and tables generated in each chunk. However, we realized we did not correctly upload the code to Zenodo previously. The new version of the code has now been uploaded and is available at the same DOI (dated September 17, 2024: [10.5281/zenodo.10892334](https://doi.org/10.5281/zenodo.10892334)).

Reviewer #2 (Remarks to the Author):

The authors have thoroughly and satisfactorily addressed this reviewer's previous comments in the revised manuscript. As mentioned previously, the manuscript rigorously assesses the relevant antibody assays for the degree of protection against dengue disease, in the context of dengue virus serotypes 2 and 3 circulation and a limited number of children previously unexposed to dengue.

A: We thank the reviewer for their helpful and thorough edits on our manuscript. The changes recommended improved the quality of our work.

I do still have a question about the asymptomatic infection determinations in the subset of the children dengue naive, with monotypic antibody and multitypic antibody. Why do the numbers tested change between the first submission and the revised manuscript (48N, 42 Mono in the first and 79N, 68 Mono in the revision)?

A: Thank you for this important question. The reviewer's initial comment prompted us to re-evaluate the number of individuals with asymptomatic infections. On review of the data, we realized that additional PRNT assays had been performed on naïve individuals since the original drafting, and these were added into the dataset. We also realized that the neutralization assays on

the monotypics were not performed on a randomly selected group, but rather were selected based on their Luminex results. In contrast, the Luminex analysis was performed on a random selection of monotypics. Thus, we felt that using the Luminex data to identify asymptomatic infection in the monotypics was preferable. As compared to the neutralization assay, there were additional individuals evaluated by Luminex and these monotypics were included in the Luminex correlates analysis.

According to Figure 3 91 or 92 asymptomatic infections were identified out of the subset of 213 individuals in the first year. This overall asymptomatic infection rate of 43% in one year is extremely high and seems inconsistent with 57 symptomatic infections ascertained after 5 years, a 4% illness incidence rate in the cohort of unvaccinated children. Do the authors think these 92 children have been aviremically boosted?

A: Thank you for this insightful question. We do feel that the asymptomatic infection rate is correct during this period of the cohort, as high seroconversion rates were observed for both vaccinated and unvaccinated subjects during a time corresponding with a major dengue epidemic¹. Additionally, as noted in the methods section (lines 130-131), enrollment occurred in May of 2017, but febrile surveillance did not begin until November 1, 2017. There is often high dengue transmission between May and November², and cases of symptomatic dengue were almost certainly missed prior to the start of febrile surveillance. In contrast, asymptomatic cases could be captured by noted changes in antibody responses. Thus, the difference in symptomatic vs asymptomatic infection rates in the first year likely reflects a longer surveillance period for asymptomatic infection, and that the majority of dengue infections are typically asymptomatic³.

Also line 157 in Methods indicates testing for asymptomatic infection was done "approximately one year after enrollment." Line 357 in Results indicates infections ascertained "over the first 1-2 years of follow-up).

A: This is a confusing discrepancy. Line 357 is correct that infections were evaluated over the first 1-2 years of follow-up. Thus, line 157 has been updated as follows:

Methods: Line 157, now line 246

“This was performed by comparing antibody levels at baseline versus those collected 1-2 years after enrollment.”

Minor editorial

1) Line 489 - Replace "it" with "is."

A: Thank you for catching this typo. Line 489 has been updated as follows:

Discussion: Line 489, now line 468

“...whether the mature neutralization assay is as effective a correlate of protection as assays that measure these distinct antibody populations separately.”

2) Suggest replacing reference 7 on TAK-003 dengue vaccine trial with the final trial

publication by Tricou et al, *Lancet Global Health* 12(2) E257-270, Feb 2024

A: This is an excellent point, and the reference replacement has been made to line 47 in the introduction.

References

1. Ylade M, Crisostomo MV, Daag JV, et al. Effect of single-dose, live, attenuated dengue vaccine in children with or without previous dengue on risk of subsequent, virologically confirmed dengue in Cebu, the Philippines: a longitudinal, prospective, population-based cohort study. *The Lancet Infectious Diseases*. March 22 2024;24:S1473-3099. doi:10.1016/S1473-3099(24)00099-9
2. Xu Z, Bambrick H, Yakob L, et al. High relative humidity might trigger the occurrence of the second seasonal peak of dengue in the Philippines. *Sci Total Environ*. Mar 15 2020;708:134849. doi:10.1016/j.scitotenv.2019.134849
3. Asish PR, Dasgupta S, Rachel G, Bagepally BS, Girish Kumar CP. Global prevalence of asymptomatic dengue infections - a systematic review and meta-analysis. *Int J Infect Dis*. Sep 2023;134:292-298. doi:10.1016/j.ijid.2023.07.010